# Sweet Orange Juice Processing By-Product Extracts: A Caries Management Alternative to Chlorhexidine

**DOI:** 10.3390/biom13111607

**Published:** 2023-11-02

**Authors:** Suvro Saha, Christine Boesch, Joanne Maycock, Simon Wood, Thuy Do

**Affiliations:** 1School of Food Science and Nutrition, Faculty of Environment, University of Leeds, Leeds LS2 9JT, UK; suvrosaha@gmail.com (S.S.);; 2School of Dentistry, Division of Oral Biology, Faculty of Medicine & Health, University of Leeds, Leeds LS9 7TF, UK

**Keywords:** citrus waste, biofilm, oral microbiome, caries, antimicrobial, chlorhexidine

## Abstract

Dental caries is one of the most prevalent chronic diseases globally in both children and adults. This study investigated the potential of industrial sweet orange waste extracts (ISOWE) as a substitute for chlorhexidine (CHX) in managing dental caries. First, the cytotoxicity of ISOWE (40, 80, 120 mg/mL) and CHX (0.1 and 0.2%) on buccal epithelial cells was determined. ISOWE exhibited no overall toxicity, whereas CHX strongly affected cell viability. The combination of ISOWE and CHX significantly enhanced cell proliferation compared to CHX alone. Next, the antimicrobial efficacy of ISOWE, CHX, and their combination was assessed against a 7-day complex biofilm model inoculated with oral samples from human volunteers. CHX exhibited indiscriminate antimicrobial action, affecting both pathogenic and health-associated oral microorganisms. ISOWE demonstrated lower antimicrobial efficacy than CHX but showed enhanced efficacy against pathogenic species while preserving the oral microbiome’s balance. When applied to a cariogenic biofilm, the combined treatment of ISOWE with 0.1% CHX showed similar efficacy to 0.2% CHX treatment alone. Overall, the findings suggest that ISOWE is a promising natural anti-cariogenic agent with lower toxicity and enhanced selectivity for pathogenic species compared to CHX.

## 1. Introduction

The human oral and gut microbiome is a vast and complex community of trillions of microbes, containing approximately 45.6 million genes, which is 2000 times more genes than the human genome [1]. Lin and Peddada [1] have referred to the microbiome as the “second genome” of the human body. Any disruption of the oral microbiome, known as dysbiosis, can lead to diseases such as tooth decay (caries) and gum disease (periodontitis) [2,3]. The oral microbiome is the second most complex bacterial community in the human body after the gut microbiome [4]. It is estimated that over 1000 different species of bacteria colonize the oral cavity, creating a diverse and delicate ecosystem [5]. A healthy oral microbiome maintains balance among these complex communities [4]. An imbalance in the oral microbiome can not only contribute to dental caries, the most common non-communicable disease worldwide, but it may also contribute to systemic diseases such as metabolic disorders, cardiovascular disease, Alzheimer’s disease, and even pregnancy complications [6]. One of the main factors contributing to dental caries is a diet high in sugar, which facilitates the overgrowth of acid-producing bacteria [7,8].

Chlorhexidine (CHX) is widely used to treat gingivitis, periodontitis, and to prevent caries, but it is associated with side effects such as reduced bacterial diversity and abundance of beneficial bacteria, which are required to maintain oral and systemic health [9,10]. Additionally, studies have shown that various bacterial taxa are developing resistance to CHX, as has been observed with other antibiotics [11,12]. An ideal product for managing oral cavity hygiene should maintain the homeostatic balance within the oral biofilm by controlling the number of pathogens without damaging beneficial bacteria or host tissues [13,14]. Therefore, there is a high global demand for organic antimicrobials that can prevent dental caries without having a deleterious impact on the oral microbiome [15].

There is an increasing interest in utilizing plant secondary metabolites such as polyphenols, many of which have demonstrated a range of biological activities including antimicrobial properties. Sweet orange is the most widely cultivated and processed citrus fruit [16], and its juice production generates a significant amount of by-product (50–60% of the original fruit) in the form of peel [17]. Sweet orange peel contains high amounts of polyphenols, especially flavonoids, which are beneficial for human health [18]. However, despite its valuable composition, sweet orange peel is often treated as industrial waste and used as animal feed or disposed of in landfills [19]. Citrus juicing waste is indeed an important production side stream and composed of mostly peel. These compounds can be sourced from edible crops as well as recovered from industrial side-streams. We have recently demonstrated the potency of flavonoid-rich industrial sweet orange waste extract to reduce the thickness and viability of cariogenic biofilm. Extraction is a key step to utilize the phytochemicals in this by-product of the juice industry. By extracting the polyphenols from sweet orange peel, we can create valuable products that promote health and reduce waste. This could lead to the development of novel anti-caries agents that are more sustainable and environmentally friendly than existing treatments.

Our previous study [20] used a 7-day cariogenic dual-species biofilm model to show that flavonoid-rich industrial sweet orange waste extract (ISOWE) effectively reduced the thickness and viability of the biofilm. In the present study, ISOWE was evaluated for its cytotoxicity using the TR146 cell line and its impact on a more realistic model system using a 7-day complex cariogenic biofilm model developed using oral samples (dental plaque, tongue biofilm, and saliva) collected from human volunteers. We aimed to evaluate the potential of ISOWE as an alternative to chlorhexidine (CHX) for dental caries management while minimizing disruption to the oral microbiome and toxicity to host cells. Specifically, we investigated the cytotoxicity of ISOWE and CHX and compared their impacts on the composition and viability of complex multi-species cariogenic biofilms developed from human oral microbiome samples. We used both total viable bacteria count (CFU/mL) and 16S rRNA sequencing to profile the effects of ISOWE, CHX, and their combination on the oral microbiota.

## 2. Materials and Methods

### 2.1. Sample Preparation and Extraction

Sweet orange juicing waste was kindly provided by The Juice Executive (Kent, UK). The details of the sample treatment and extraction process were mentioned in our previous study [20].

### 2.2. Preparation of Treatment Solutions with ISOWE, CHX, and Combination of CHX and ISOWE

Stock solutions of ISOWE (250 mg/mL) and 2% CHX (Merck Life Science UK Ltd., Gillingham, UK) were prepared for both cytotoxic and anti-cariogenic assays, using 4 and 2% DMSO (Sigma-Aldrich Ltd., Dorset, UK), respectively, and further diluted to different concentrations of CHX (0.1 and 0.2%) and ISOWE (40, 80, 120 mg/mL). Gibco Dulbecco’s Modified Eagle Medium (DMEM, ThermoFischer Scientific, Oxford, UK) was supplemented with 10% Fetal bovine serum (FBS, Sigma-Aldrich, Dorset, UK) and 1% Penicillin/Streptomycin (P/S, Thermofisher Scientific, Loughborough, UK) for cytotoxicity assays. PBS (Thermofisher Scientific, Loughborough, UK) was used to carry out anti-cariogenic assays against the complex biofilm model. Equal volumes of each concentration of CHX and ISOWE were mixed for combination treatments. The sugar content and pH of the extracts were, however, not measured.

### 2.3. Cytotoxic Assays

#### 2.3.1. Cultivation of TR146 Cells

The human buccal carcinoma cell line TR146 was kindly provided by Prof Francisco M. Goycoolea, School of Food Science and Nutrition. The cells were routinely cultivated in DMEM medium with experiments being conducted within five passages.

#### 2.3.2. Cell Viability and Cell Proliferation Assays

Both cell viability and proliferation were determined via MTT assay. The experimental design was based on the study conducted by Lessa et al. [21] with modifications. TR146 cells were seeded (0.02 × 10^6^ cells/mL) in 48-well plates at a volume of 500 µL per well. The plates were incubated at 37 °C and 5% CO_2_ throughout the experiment. The medium was changed every two days until the confluence reached 70%. The cells were then treated with CHX (0.1 and 0.2%), ISOWE (40, 80, and 120 mg/mL), and combinations of each concentration of CHX and ISOWE, prepared in complete DMEM. The experiments were conducted in triplicate.

The cell viability assays were performed immediately after treating the cells with one dose (1D_Via) and two doses (2D_Via) of the respective solutions. In the case of 1D_Via, the media was aspirated, and the wells were washed with 1x DPBS. A total of 500 µL of the respective treatment solutions was then added to the designated wells for 60 s. After the aspiration of the treatment solutions, the wells were washed again with 1× DPBS before the MTT assay was performed. The 2D_Via cells were treated twice for 60 s, with a 12-h interval between treatments. After the first treatment, the wells were washed, and 500 µL of fresh medium was added to each well. After a 12-h incubation, the cells were treated a second time in the same way and then processed for the MTT assay as described above. The cell proliferation assay was performed by allowing the cells to grow for 24 h after the last treatment and then assaying the percentage of cell viability using the MTT assay. The cells were treated in the same way as for the viability assay. After the final washing step, new medium was added to the wells and the plates were incubated for 24 h to allow the cells to proliferate. The wells were then washed again and processed for the MTT assay. The proliferation assay samples after one dose and two doses were labeled 1D_Pro and 2D_Pro, respectively.

#### 2.3.3. MTT Assay

After treatments, cells in 24-well plates were incubated with 1 mM MTT working solution in serum-free DMEM and incubated with the dye at 37 °C for 4 h. Blank samples were prepared by adding MTT dye solution to empty wells. The dye-containing medium was then aspirated, the cells washed with 1x DPBS and 500 µL of DMSO added to each well to release the dye. The plate was kept on the plate shaker for 15 min and the DMSO solution from each well was then transferred into two wells (200 µL/well) of a 96 well plate to measure the absorbance at 540 nm using TECAN (Mannedorf, Switzerland). Because the dye is light sensitive, the assays were carried out under light protected conditions. The mean OD of the blank was subtracted from the mean of each treatment group. The mean OD of the untreated control cells (where cells were treated with only media) was set to represent 100% viability of the respective experimental group. The results were expressed as viability %.

### 2.4. Seven-Day Complex Cariogenic Biofilm Model Development and Anti-Cariogenic Assays

#### 2.4.1. Volunteer Recruitment and Sample Collection

Ethics approval for sample collection was granted by the University of Leeds Dental School Research Ethics Committee (160420/SS/298). Fifteen volunteers (10 females and 5 males) were recruited to provide samples of biofilm from the tongue dorsum, 2–3 mL of saliva, and dental plaque from between teeth and between the tooth and gum line. The exclusion criteria for volunteers were the following: not diagnosed with any systemic diseases such as cardiovascular diseases, diabetes mellitus, not used antibiotics for at least 3 months before sample donation, not undergone oral treatment including surgeries for at least 3 months before sample donation, neither pregnant nor lactating woman. Participants were asked to refrain from practicing any kind of oral hygiene (including teeth brushing and/or using any mouthwash) for a minimum of 12 h before collecting the saliva, plaque, and tongue scraping sample.

#### 2.4.2. Sample Processing and Development of the 7-Day Complex Cariogenic Biofilm Model

The artificial saliva used in the biofilm-growth medium (BGM) was a modified version of the artificial saliva described in Saha et al. (2023) [20]. The BGM was prepared by diluting artificial saliva (600 mL/L) with basal media (200 mL/L) following the addition of heat inactivated fetal bovine serum (200 mL/L) to simulate the gingival environment as described by Naginyte [22]. Sucrose (2%) was added to the basal media to prepare sucrose fortified BGM (SEBGM).

All the sample tubes were placed in the anaerobic workstation immediately after being received from each volunteer and the entire experiment was conducted under the same anaerobic conditions (10% CO_2_, 10% H, and 80% N) at 37 °C. All the saliva samples were mixed in a sterile 50 mL centrifuge tube and vortexed for 1 min to obtain an even distribution of microorganisms. The same process was performed with plaque and tongue scraping samples.

The complex cariogenic model is based on our previous dual-species cariogenic model [20]. In a sterile Eppendorf tube, a sterile hydroxyapatite (HA) disc was kept in a vertical position, and 100 µL of each oral sample was pipetted onto it followed by gentle mixing and incubation for 24 h. After the incubation period, the HA disc was gently washed 3 times with PBS to remove loose microbial cells. The development of the complex cariogenic biofilm model can be briefly summarised as follows. After washing the disc with PBS, 200 µL of SEBGM was added to the Eppendorf tube to conduct sucrose exposure. After 20 min, the HA disc was washed with PBS and incubated in BGM. Similarly to the dual-species cariogenic model, the sucrose exposure was conducted 3 times a day at 6 h-interval, continued for 7 days to form a cariogenic biofilm. In this experiment, there were two groups of controls: biofilm development without sucrose exposure (Biofilm without sucrose) and cariogenic biofilm development with sucrose exposure (Biofilm with sucrose).

After developing 7-day mature complex cariogenic biofilms on HA discs, we treated them for 4 days. Each day, we washed the HA discs with PBS, administered the first dose of the treatment for 1 min, washed them again with PBS, and incubated them in BGM. We administered the second dose on the same day, after 12 h, following the same procedure. We exposed both control groups to PBS instead of the treatment solution. On the day following the treatment, we assayed anticariogenicity by measuring the viable bacterial count (CFU/mL) and processing samples for 16S rRNA amplicon sequencing. We conducted all experiments in triplicate.

#### 2.4.3. Viable Bacterial Count (CFU/mL)

Both brain–heart infusion (BHI) and Columbia blood agar (CBA) plates were used to determine the viable bacterial count in samples from the complex oral biofilm model. After making 10-fold serial dilutions (10–1 to 10–10) from the biofilm solution, 100 µL of each dilution was spread in duplicate onto both BHA and CBA agar plates. The plates were then incubated for 72 h anaerobically at 37 °C and the colonies were counted.

##### DNA Extraction of Bacteria from the 7-Day Complex Biofilm

DNA was extracted using QIAamp PowerFecal Pro DNA kit (Qiagen, Hilden, Germany) according to the manufacturer’s instructions. The extracted DNA was stored at −20 °C until processing for 16S rRNA sequencing. The total DNA quality and quantity were determined by a TECAN microplate reader using NanoQuant Plate (Mannedorf, Switzerland). The purity was evaluated as the ratio of absorbance at 260 and 280 nm.

##### Amplicon Sequencing Outsourced to Novogene UK Ltd.

The HA discs were washed with DNase/RNase-free distilled water (Invitrogen, Oxford, UK) to remove planktonic bacteria. They were then transferred to another sterile Eppendorf tube containing 1 mL of nucleic acid protecting reagent (Qiagen, Manchester, UK) with 3 to 4 sterile glass beads. The HA discs were vortexed for 1 min to collect the bacterial load from the HA disc into the protecting reagent followed by transferring the biofilm solution in another sterile Eppendorf tube. It was stored at −20 °C until processed further. Purified DNA was further processed by Novogene Company Limited (Cambridge, UK) for amplicon sequencing. The V3–V4 region of the hypervariable region of the 16S rRNA gene was amplified [23,24], and equimolar amount of DNA from each sample was pooled. These pooled amplicons were subjected to end-repair, A-tailing, and ligation with an Illumina adapter to construct the DNA library. The quality of the libraries was checked using Qubit and Bioanalyzer. Finally, the libraries were sequenced on the Miseq Illumina platform to generate 250 bp paired-end reads.

The paired-end reads from the original DNA fragments were merged using FLASH (Fast Length Adjustment of SHort reads). Paired-end reads were assigned to each sample based on the unique barcode of each sample. QIIME2 software (version 2022.2) was used to denoise the sequence by filtering out the sequence with less than 5 abundances to generate the final ASV (Amplicon Sequence Variables). QIIME2 was used to calculate alpha diversity and beta diversity. The alpha diversity indices included Chao1 and Shannon.

### 2.5. Statistical Analysis

The data were reported as mean ± standard deviation and a significance threshold of *p* < 0.05 was applied. GraphPad Prism version 9.3.1 was used to plot the graphs for assessment of viability and proliferation, bacterial viable count (CFU/mL), alpha indices, and taxonomic profile, including, phyla, genus, and species. Python v.3.9.5 was used to plot the graphs for beta diversity including PCA. All the data were analyzed using ANOVA, with subsequent Tukey’s multiple comparisons test being used to detect significant differences (*p* < 0.05) between the experimental groups.

## 3. Results

### 3.1. Effect of CHX and ISOWE on TR146 Cell Viability and Proliferation

Experiments were considered either as non-toxic, weakly toxic, moderately toxic, or strongly toxic, based on the percentage of the cell viabilities being above 80%, within the range 80–60%, 60–40%, and below 40%, respectively [25]. Figure 1a shows that none of the ISOWE concentrations (120, 80, and 40 mg/mL) exhibited toxicity after a single dose (1D_Via) application. Although 80 and 40 mg/mL of ISOWE showed more than 90% cell viability, there was a reduction of cell viability (85.10%) at the highest concentration of ISOWE with 120 mg/mL. The viability was not significantly different between 80 and 40 mg/mL of ISOWE. The cytotoxicity of CHX was dose-dependent. At a single time, exposures of 0.2 and 0.1% CHX were, respectively, moderately and weakly toxic (55.39 and 63.07% cell viability, respectively). There was no significant difference in the percentage of cell viability for the combinations of each concentration of ISOWE and 0.2% CHX, and the same trend was seen for the combinations with 0.1% CHX. The combinations with 0.1% CHX showed less toxicity compared to the combinations with 0.2% CHX. Overall, our findings suggest that ISOWE is non-toxic at all concentrations tested, while CHX is moderately toxic at a concentration of 0.2% and weakly toxic at a concentration of 0.1%. Combining ISOWE and CHX did not increase toxicity more than either treatment alone.

In the 2D_Via experiments, the cell viability was less than 60% after immediate exposure to both CHX concentrations. The percentage of cell viability values were 90.5, 89.9, and 79.7% for 40, 80, and 120 mg/mL of ISOWE, respectively (Figure 1b). Unlike in 1D_Via, cell viability was significantly increased when the TR146 cells were in the presence of the combined solution of ISOWE and 0.1% CHX, compared to 0.1 and 0.2% CHX alone.

Both single and double doses of CHX significantly reduced cell proliferation. The double dose caused the percentage of cell viability to decrease below 40%. The ISOWE did not have any significant effect on cell proliferation. However, the combinations of ISOWE with CHX showed improvement in the cell proliferation compared to the respective CHX treatment.

In the case of cell proliferation assays, the double dose of 120 mg/mL of ISOWE resulted in a ~14% increase in cell viability. The other two concentrations did not show a significant difference compared to the control. Unlike the impact of the single dose on cell proliferation, the double dose with the CHX (0.2 and 0.1%) and ISOWE (80 and 40 mg/mL) did not show a significant difference. However, the combination of 0.1% CHX and 120 mg/mL ISOWE showed a significant increase in cell viability, indicating that this combination was able to reduce the cytotoxicity of 0.1% CHX, but the same ISOWE concentration was not able to reduce the cytotoxicity of 0.2% CHX.

### 3.2. Viable Bacterial Count (CFU/mL)

The bacteriostatic/bactericidal property of 120 mg/mL ISOWE was explored using a complex biofilm model, as this concentration did not show any negative impact on the cell viability and proliferation of TR146 cells. In combination with 0.1% CHX, ISOWE showed less cytotoxicity than 0.1% CHX alone, and it also showed better anti-biofilm activity than 0.2% CHX in our previous work using a dual-species biofilm model. Figure 2 indicates the viable bacterial count (CFU/mL) for the complex cariogenic model on BHI and CBA microbial growth media.

In both media, biofilms with sucrose supplementation (cariogenic biofilms) had significantly higher colony forming unit (CFU) counts than biofilms grown in the absence of sucrose (non-cariogenic biofilms). However, the viable count for both non-cariogenic and cariogenic biofilms was significantly higher on BHI agar (42 − 49 × 107 and 11 − 14 × 1010, respectively) than on CBA (25 − 31 × 107 and 85 − 91 × 109, respectively). The bacterial viability was inversely proportional to CHX concentration. Although the bacterial viability count for 0.1% CHX did not show any significant difference between the two growth media, in the case of 0.2% CHX treatment, the CFU/mL count for CBA (14 − 22 × 107) was significantly lower compared to the count on BHI media (28 − 38 × 107). The ISOWE significantly reduced the bacterial viability compared to the control cariogenic biofilm. The CFU/mL count for ISOWE treatment on BHI media (19 − 25 × 108) was significantly lower compared to the viability count on CBA media (32 − 36 × 108). The anticariogenicity of CHX was higher compared to ISOWE, but the combination treatment (0.1% CHX and ISOWE) was more effective compared to treatment with 0.2% CHX alone. There was no significant difference in the CFU/mL count between combination treatment and non-cariogenic biofilm according to growth on CBA media. However, the CFU/mL count on BHI media for the combination treatment was significantly lower compared to non-cariogenic biofilm.

### 3.3. Alpha and Beta Diversity

Alpha diversity is a measure of the species diversity (or richness) within a microbial community, and in this study, it was estimated based on the Chao1 and Shannon index (Figure 3).

Both diversity indices were significantly higher for the cariogenic biofilm compared to the non-cariogenic biofilm, as well as other treatment parameters. The dose-dependent antimicrobial activity of CHX was established in both measured alpha diversity indices. Chao1 was significantly lower and higher for ISOWE-treated cariogenic biofilm and control non-cariogenic biofilm, respectively. The statistical trends for both Shannon and Chao1 diversity indices for ISOWE-treated cariogenic biofilm were the same. The Chao1 index for cariogenic biofilm treated with a combination of 0.1% CHX and ISOWE was significantly lower compared to treatment with 0.2% CHX alone. However, the Shannon diversity index did not show any significant difference between 0.2% CHX treatment alone and combination treatment. Both diversity indices were significantly lower for the combination treatment compared to a non-cariogenic biofilm.

The beta diversity is a measure of the similarity or dissimilarity between two samples or communities. The distance between samples was calculated using principal coordinates analysis (PCoA) to explain phylogenetic variation based on the unweighted UniFrac values (Figure 4).

The plot shows that five major clusters formed from six experimental parameters, which are biofilm without sucrose exposure, cariogenic biofilm (biofilm with sucrose exposure), and effect of CHX (0.1 and 0.2%), ISOWE, and combination of 0.1% CHX and ISOWE treatment on the complex cariogenic model. The microorganisms detected in the combination of 0.1% CHX with ISOWE and 0.2% CHX-treated cariogenic biofilm alone, are closely related as they were clustered together. Apart from these two treatment groups, the PCoA plot suggested that grouping by other treatment groups was statistically significant.

### 3.4. Analysis of Bacterial Composition on Phylum, Genus, and Species Levels

In all experimental groups, bacteria were found to be predominant compared to the remaining two detected kingdoms (unassigned and archaea) (Table 1). Overall, a total of 588 bacterial species were detected from all experimental parameters, including biofilm without sucrose (197 species), cariogenic biofilm (358 species), and cariogenic biofilm treated with 0.1 and 0.2% CHX (167 and 123 species, respectively), ISOWE (181 species), and combination of 0.1% CHX with ISOWE (131 species). In Table 2, the percentage of relative abundance for unassigned groups in each bacterial taxonomical rank for each sample group was shown.

All these detected species belong to 15 phyla (Figure 5) and 179 genera. The identified bacterial phyla included Firmicutes, Spirochaetota, Bacteroidota, Actinobacteriota, Campilobacterota, Synergistota, Proteobacteria, Fusobacteriota, Desulfobacteriota, Verrucomicrobiota, and Patescibacteria. In Figure 6, the 15 most abundant genera within the experimental group and the effect on the difference in relative abundance (%) for a few selective species were represented. Among the identified genus, *Lactobacillus* was predominant. According to the selected species, sucrose supplementation raised the bacterial growth in the biofilm except for *S. salivarius*. In a dose-dependent manner, CHX was more effective at reducing biofilm growth compared to ISOWE. The combination of ISOWE with 0.1% CHX treatment has been able to further reduce the abundance of every species in the cariogenic biofilm compared to treatment with 0.2% CHX alone.

## 4. Discussion

An ideal antiplaque agent should be effective against microbes and safe for patients [13]. Therefore, it is important to assess the dose-dependent toxicity of all compounds used in caries management to determine their individual safety thresholds. In vitro cytotoxicity assessment is a useful initial biological screening tool because it is fast, reproducible, controllable, and cost-effective [26]. Cell culture is a common method for testing the toxicity of oral healthcare products [27]. In this study, both concentrations of CHX reduced cell viability and proliferation, and the toxicity was proportional to concentration and treatment duration. Müller et al. [28] evaluated the cytotoxicity of 12 commercially available mouthwashes on primary human oral fibroblast cells with 2 min of contact time. The mouthwash containing 0.05% CHX was less toxic (lethal-dose concentration LD50 > 60%) than the mouthwash containing 0.2% CHX (LD50 < 10%). Other studies have also reported that CHX toxicity increased with higher concentration and treatment duration [21,29,30]. However, the studies used different cell lines, CHX dosages, and cytotoxicity test methods.

Previous studies have shown that CHX can be toxic to oral cells, even at low concentrations. Lessa et al. [21] found that all CHX concentrations ranging from 0.06 to 2.0% were toxic to MDPC-23 cells after 2 h of exposure. Yayli et al. [29] and Emmadi et al. [31] reported that 0.2% CHX can significantly reduce the viability and proliferation of human gingival fibroblast cells (HGFs), even after short exposure times. Wyganowska-Swiatkowska et al. [32] observed that 0.2% CHX treatment caused HGF cells to become smaller and rounded. Both Lessa et al. [21] and Chang et al. [33] found that 0.2% CHX can inhibit protein synthesis. The present study found that CHX was less toxic to host cells than reported in some previous studies. This may be due to differences in the cell lines used, the exposure times, or the methods used to assess toxicity.

Polyphenols are plant compounds that have a variety of health benefits, including antioxidant and anti-inflammatory effects. They can also bind to oral surfaces, such as the tongue, taste buds, buccal mucosa, and teeth. This allows them to persist in the mouth for a prolonged period of time after consumption [34], up to several hours. This slow release gives it more time to interact with the oral cells and reinforces the antimicrobial response in the oral cavity. The interaction between polyphenols and oral cells is determined by the concentration of polyphenols and the duration of exposure.

In addition to their antimicrobial effects, polyphenols also act as antioxidants. This can help to protect oral tissues from damage caused by chronic inflammatory conditions such as periodontal disease. Edible plants typically contain β-glycosylated flavonoids [35], which can act as glucose substrates for oral bacteria and therefore can directly contribute to cell proliferation [36]. In the oral cavity, glycosylated flavonoids are hydrolyzed into their respective aglycones by an enzyme called β-glucosidase enzymes [37]. The sources of this enzyme are oral epithelial cells and oral commensals [37,38]. After hydrolysis, the aglycones are readily absorbed into oral epithelial cells. Walle et al. [37] have reported a 60% loss of the glycoside form of quercetin after 5 min of holding of a solution in the mouth, and approximately 40% of the degraded quercetin was absorbed in epithelial cells. The rate of hydrolysis of quercetin is affected by the sample matrix, as enzymes have less access to flavonoids in solid matrices, resulting in lower absorption and reduced impact on oral health [37]. In contrast, enzymes can more readily reach flavonoids in liquid matrices, such as mouthwash, leading to higher absorption and a greater impact on oral health. However, some herbal mouthwashes contain chlorhexidine (CHX), which can be toxic to oral cells [29,36,39].

A study by Ali et al. [40] investigated the cytotoxicity of citrus peel extracts (*C. sinensis* and *C. limon*) and showed a negative impact on the onion bulb roots, with *C. sinensis* having a greater impact than C. limon. The cytotoxicity was dose-dependent, but the experiment was conducted with a long treatment duration (72 h). This means that the findings may not be relevant to caries-related solutions, which are typically used for a short duration (30 s to 2 min) [41]. Mandal et al. [42] conducted a clinical trial in which human volunteers with severe gingivitis rinsed their mouths twice a day with an extract of orange peel for 14 days. The extract was effective at reducing plaque and gum inflammation, more so than a CHX rinse. This suggests that orange peel extract has stronger anti-inflammatory properties than CHX, which is a commonly used mouthwash. The researchers attributed this activity to the phenolic compounds and flavonoids in orange peel extract.

In the present study, Orange extracts counteracted the antiproliferative effects of CHX in a dose-dependent manner. Iswariya et al. [43] reported similar findings, showing that pulp extracts of *Citrus limetta* and *Citrus sinensis* improved H_2_O_2_ toxicity in yeast and leukemic cells. However, the pulp extracts were more toxic to leukemic cells than to yeast cells.

The Dutch microbiologist Lourens Gerhard Marinus Baas Becking proposed the hypothesis that all types of microorganisms, including both pathogenic and non-pathogenic species, are present in all environments, but the environmental conditions determine which types of microorganisms are able to thrive [44]. The ecological plaque hypothesis of caries development builds on this concept, suggesting that an imbalance in the oral microbiota caused by environmental stress, such as sucrose consumption, can lead to the overgrowth of cariogenic bacteria [45].

A viable count assay using two different nonselective media (BHI and CBA) showed that ISOWE has antimicrobial activity against cariogenic biofilms. Using two different nonselective media can provide information on the impact of ISOWE on a diverse range of microorganisms, as bacterial growth rate and type are influenced by nutrient availability. However, the findings varied between the two media due to the influential effect of culture media on bacterial growth rate [46,47]. Additionally, slow colony-forming bacteria may be overlooked, as they may not be able to compete with fast-growing bacteria for nutrients during the incubation period. Therefore, the CFU method may not be sufficient for mixed culture experiments, due to the variation in bacterial replication rates and taxon-specific differences [48,49]. Another disadvantage of this method for mixed culture biofilms is that it does not allow for species-level identification. However, the CFU method is widely used to assess the antimicrobial properties of substances due to its simplicity and low cost [50].

Next-generation sequencing technologies, such as 16S rRNA gene sequencing, can be used to overcome the limitations of assessing antimicrobial efficacy against complex cariogenic models [49,51]. In this study, the V3–V4 regions were sequenced, as this region is considered to be reliable for the analysis of the oral microbiome [24,52].

Similar predominant bacterial species were reported in the present study to those reported by Baumgardner et al. [53] and Modafer et al. [54]. A small number of archaea were also detected. The relative abundance of archaea was increased in sucrose-supplemented complex (cariogenic) biofilms, similar to bacteria. Although archaea can be present in healthy oral biofilms, an increased number and prevalence of archaea in the oral biofilm has been reported for various infectious diseases, including gastrointestinal and periodontal diseases [4,55]. The species that belong to this kingdom are called methanogens. *Methanobrevibacter smithii* and *Methanobrevibacter oralis* were detected in this study. *M. oralis* is specifically linked with periodontitis [55,56]. Yamabe et al. [57] reported that *M. oralis* was detected through IgG antibodies in 70% of acute periodontitis patients but not in healthy volunteers. This species has also been reported to show antimicrobial resistance to tetracycline [58]. In the CHX- and CHX + ISOWE-treated cariogenic biofilms, both identified methanogenic archaea species were not detected. In ISOWE-treated cariogenic biofilm, *M. oralis* was detected in lower abundance compared to the cariogenic complex biofilm, but the other methanogenic archaea species was not detected.

Firmicutes were the most abundant phylum in this study, consistent with previous findings [52,59,60]. Li et al. [61] reported that Actinobacteriota was the most abundant phylum in their study, followed by Firmicutes. The composition of the oral microbiota can vary with diet, geography, and climate [62]. In the presence of sucrose in the biofilm, the abundance of Firmicutes (mainly *Lactobacillus* and *Streptococcus*) increased significantly. In another study, *Streptococcus* was reported to be the most prevalent Firmicutes phylum [60]. When sugar is abundant in the oral cavity, Firmicutes thrive [52]. The growth of Firmicutes species, such as *Streptococcus mutans* and *Lactobacillus casei*, is proportional to the synthesis of lactic acid as a metabolic byproduct of sugar metabolism. This leads to a drop in the pH of the oral biofilm, resulting in the development of cariogenic biofilm. Except for Spirochaetes, the growth of all other detected phyla was increased in the cariogenic biofilm compared to the biofilm developed in the absence of sucrose. *Bifidobacterium*, a member of the Actinobacteriota phylum that has been reported to be present at high levels in the saliva of caries patients, was one of the 15 most abundant phyla in this study [63].

*Corynebacterium*, another genus in the Actinobacteriota phylum, was detected in this study. *Corynebacterium* helps in the attachment of late-colonizing pathogens, leading to the formation of mature cariogenic biofilm [64]. Studies on anticariogenic approaches have proposed specifically targeting *Corynebacterium* to inhibit the development of mature cariogenic biofilm [65,66,67]. The abundance of both *Bifidobacterium* and *Corynebacterium* was efficiently reduced with ISOWE treatment on the complex cariogenic model.

Similarly to *Corynebacterium*, *Fusobacterium* acts as a connective bridge between early and late plaque colonizers. It was detected in the biofilm with and without sucrose, and in the ISOWE-treated complex cariogenic model, but at a lower level than in the cariogenic biofilm.

The core microbiome (least variable) of a healthy oral cavity includes Firmicutes, Actinobacteria, Fusobacteria, Bacteroidetes, and Spirochaetes [52,59,60,68]. However, many of the microbial members from this healthy core oral microbiome are strongly linked to caries and other diseases. Along with Firmicutes, Actinobacteriota and Proteobacteria have been reported to be associated with obesity [69]. In another study, poor oral health was considered as a risk factor for depression and anxiety as an unbalanced oral biofilm can cause the synthesis of a higher level of stress hormones [70]. In Simpson et al. [70] Spirochaetes, Firmicutes, and Actinobacteria were reported to be linked with mental health. *S. mutans*, which is a member of Firmicutes, has been reported as one of the most important cariogenic species [7,8]. Despite reports suggesting the absence of this species in cariogenic biofilm, this organism has been detected in health-associated biofilms [71]. Therefore, the abundance of potential pathogenic groups of microorganisms is more important than the presence of a particular group(s) of microorganisms for caries development.

Chlorhexidine (CHX) treatment caused a significant dose-dependent reduction in the abundance of all detected taxonomic profiles in the cariogenic biofilm. This indiscriminate antimicrobial activity of CHX has been reported in other studies [10,72,73,74]. In contrast, Tribble et al. [75] reported that CHX boosts the growth of Bacteroidetes, a phylum that includes species that cause periodontal disease [76]. CHX was also found to be very effective against *Streptococcus salivarius*, a non-cariogenic species that helps to reduce plaque accumulation and acidification [77]. Burton et al. [77] reported that *S. salivarius* produces bacteriocins that can reduce the salivary load of *Streptococcus mutans* and lactobacilli. In patients with caries, the level of *Streptococcus sobrinus* decreases as the levels of pathogenic species such as Streptococcus mitis and *Streptococcus oralis* increase [78]. In this study, ISOWE was less effective than CHX against *S. salivarius*.

Brookes et al. [10] and Hyde et al. [79] have questioned the beneficial effects of CHX due to its harsh and indiscriminate antimicrobial activity on the entire oral microbiome. The oral biofilm in healthy individuals contributes to overall well-being by restricting the growth of pathogens, providing innate immunity [80,81]. The eradication of oral commensals can jeopardize oral and systemic health. In this study, CHX treatment caused a significant reduction in the number of members of the genus Prevotella, which includes nitrate-reducing species. Species in this group convert salivary inorganic nitrate to nitrite, which helps to maintain oral pH, inhibit platelet aggregation, and maintain pulmonary vascular health [10,82,83]. Using CHX mouthwash for 7 consecutive days can decrease salivary pH and buffering capacity, which can lead to cariogenic biofilm development [10,73]. CHX can decrease the species diversity and richness of the oral microbiome, while ISOWE has shown antimicrobial activity against cariogenic species and other Gram-negative oral pathogens, such as *Porphyromonas gingivalis* and *Fusobacterium nucleatum*, which are associated with periodontal diseases and caries progression [22,84,85]. The findings of this study suggest that ISOWE may have potential for managing dental caries and periodontal diseases and could be used as an adjunct in mouthwashes or other oral care products.

ISOWE effectively reduced the bacterial abundance in cariogenic biofilms, but not as harshly as CHX. However, the combination of ISOWE with 0.1% CHX was as effective as 0.2% CHX on cariogenic biofilms. This suggests a synergistic antimicrobial activity between ISOWE and CHX. Therefore, the combination of ISOWE and lower concentrations of CHX should be further investigated, as it could reduce the side effects associated with CHX while maintaining its antimicrobial strength.

No other study has reported the impact of ISOWE on the comprehensive composition of oral microbiota. One limitation of this study is that it used a culture-based biofilm model, which cannot capture the full diversity of the oral microbiome, as many oral microbes are unculturable in a laboratory setting. Additionally, 16S rRNA sequencing, which was used to identify microbes in this study, cannot provide detailed information on eukaryotic microorganisms (fungi, protozoa) and viruses that are also present in the oral microbiome [86]. While bacteria predominate the oral microbiome, some fungal species, such as *Candida* spp., *Saccharomyces* spp., *Fusarium* spp., and *Cryptococcus* spp., are also part of a healthy oral microbiota [86,87]. The presence of viruses, such as herpesviruses and papillomaviruses, in the oral microbiota is often linked with disease [86]. However, this study does not provide much information on the other members of the oral microbiota other than bacteria. It would be interesting to investigate the effect of ISOWE alone and in combination with CHX on actual patients with dental caries and periodontitis.

## 5. Conclusions

A diet high in sugar can shift the balance of the oral microbiome towards a cariogenic ecosystem by providing optimal growth conditions for caries-causing pathogens, such as *Streptococcus mutans* and *Lactobacillus* spp., which can withstand the lower pH environment of the mouth. These pathogens can eventually outnumber the oral commensals, such as *Streptococcus salivarius*, which help maintain oral health. Chlorhexidine (CHX) is a common antimicrobial agent that can kill both pathogenic and non-pathogenic bacteria. However, CHX can also have cytotoxic effects on healthy gum cells. ISOWE, a citrus extract, has also been shown to have antimicrobial properties against caries-causing pathogens, but with less cytotoxicity to healthy gum cells than CHX.

The combination of 0.1% CHX and ISOWE (120 mg/mL) was found to be as effective as 0.2% CHX alone against cariogenic biofilm models. These findings suggest that combining ISOWE and a lower concentration of CHX could be a more effective and less harmful way to treat caries. Further studies are needed to confirm these findings and to investigate the potential of ISOWE for caries prevention.

## Figures and Tables

**Figure 1 biomolecules-13-01607-f001:**
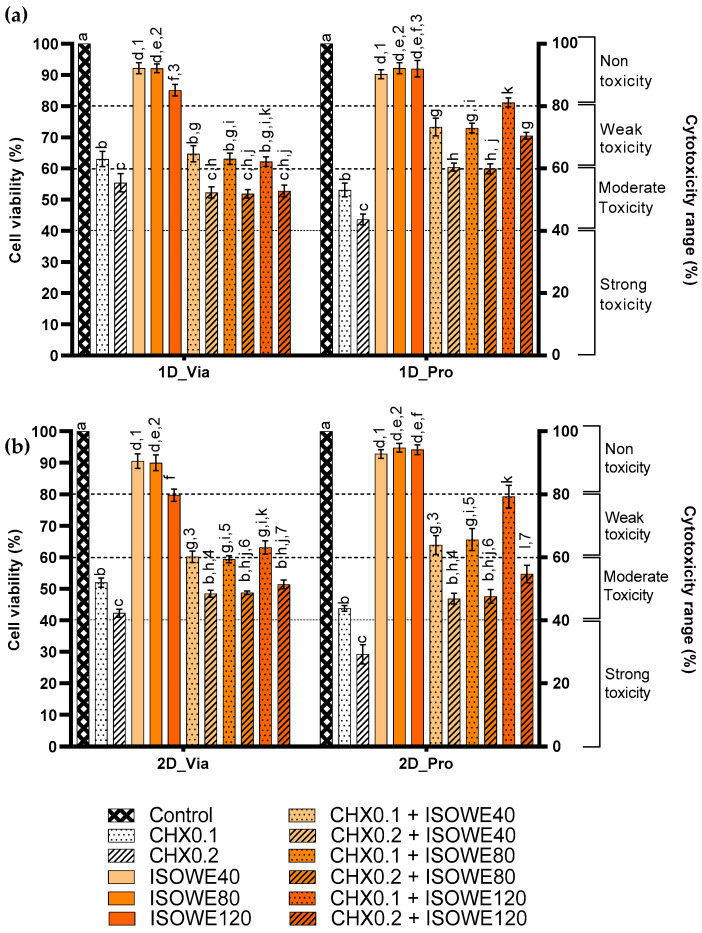
Viability and proliferation of TR146 cells after one (**a**) or two (**b**) doses of CHX and ISOWE, alone or combined. The mean OD540 of the untreated cells (Control) was considered as 100% viability. The cytotoxicity from each treatment was quantified as the percentage of cell viability relative to the untreated controls. Different letters indicate significant differences within each experimental condition. Bars that do not share similar letters indicate statistically significant differences (*p* < 0.05) and the bars sharing similar letter(s) indicate no significant difference (*p* > 0.05). Different numbers denote the significance difference (*p* < 0.05) of cell viability between an immediate and after-24-h-of-a-respective-treatment solution, and in this graph, only non-significant (*p* > 0.05) groups are marked. CHX0.1 and CHX0.2: CHX 0.1 and 0.2%, respectively; ISOWE40, ISOWE80, and ISOWE120: 40, 80, and 120 mg/mL of ISOWE; ‘+’: combination of the respective treatment solutions.

**Figure 2 biomolecules-13-01607-f002:**
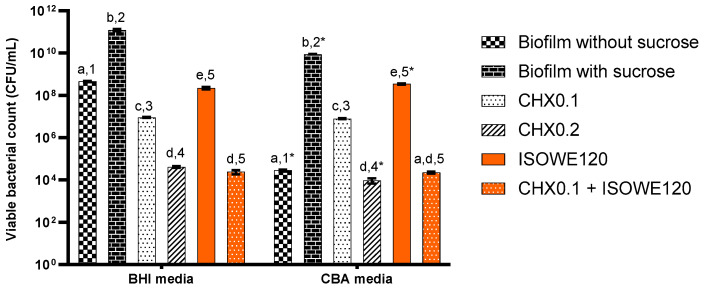
Viable bacterial count (CFU/mL) using BHI and CBA media. Biofilm samples were exposed to CHX (0.1 and 0.2%) and ISOWE, alone and in combination. Different letters indicate significant differences in CFU/mL count within each medium condition. Bars that do not share similar letters indicate a statistically significant difference (*p* < 0.05), and the bars sharing similar letter(s) indicate no significant difference (*p* > 0.05). The numbers denote the significant difference between both media for the respective treatments. The same number(s) with a “*” shows significance difference (*p* < 0.05). CHX0.1 and CHX0.2: CHX 0.1 and 2%, respectively; ISOWE120: 120 mg/mL ISOWE; and CHX0.1 + ISOWE120: CHX 0.1% with ISOWE 120mg/mL.

**Figure 3 biomolecules-13-01607-f003:**
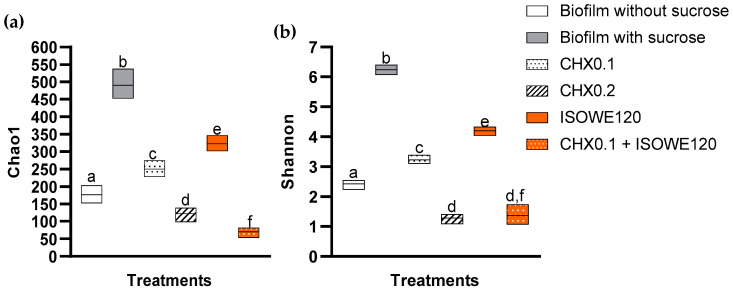
Alpha diversity indices for the 7-day complex cariogenic model. The indices were plotted with two alpha diversity indicators: Chao1 (**a**) and Shannon (**b**). The experimental parameters include biofilm without sugar, biofilm with sugar (cariogenic biofilm), and effect of CHX and ISOWE, alone and in combination. Bars that do not share similar letters indicate statistically significant differences (*p* < 0.05), and the bars sharing similar letter(s) indicate no significant difference (*p* > 0.05).

**Figure 4 biomolecules-13-01607-f004:**
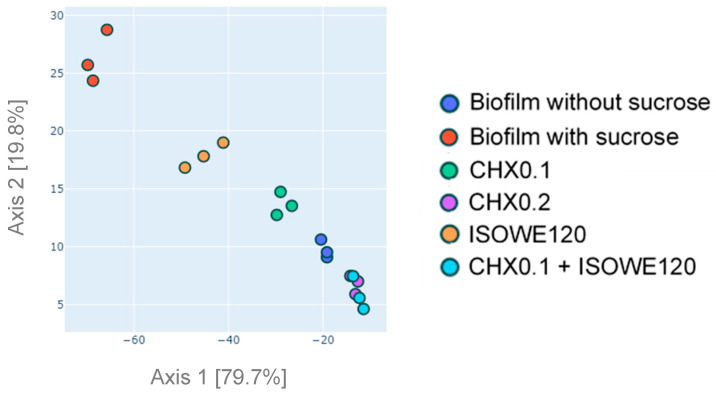
Principal component analysis (PCoA) plot. Samples were clustered based on unweighted UniFrac.

**Figure 5 biomolecules-13-01607-f005:**
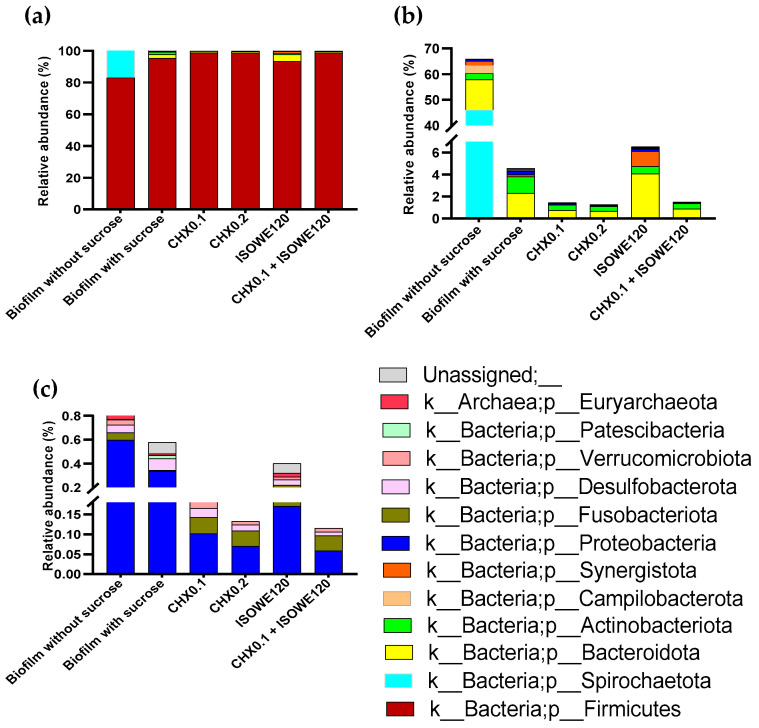
Summary of taxonomic profiles at phylum level in all experimental groups. Each color represents the average relative abundance (%) for phyla within each experimental group. The relative abundance (%) of all the identified phyla (**a**), excluding Firmicutes (**b**), and only including Proteobacteria, Fusobacteriota, Desulfobacterota, Verrucomicrobiota, Patescibacteria, and unassigned (**c**), were plotted. In legends, CHX0.1 and CHX0.2: Chlorhexidine 0.1 and 0.2%, respectively; ISOWE120: 120 mg/mL ISOWE; and CHX0.1 + ISOWE120: CHX 0.1% with ISOWE 120 mg/mL.

**Figure 6 biomolecules-13-01607-f006:**
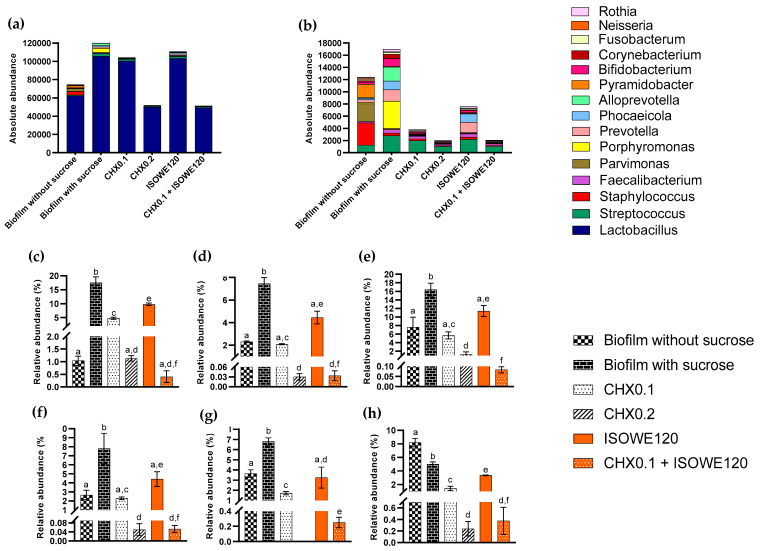
Summary of taxonomic profiles at genus (**a**,**b**) and species (**c**–**h**) level in all the experimental groups. Each color represents the average of the 15 most abundant genera within each experimental group. The identified 15 most abundant genera (**a**), and excluding *Lactobacillus* (**b**), were shown. The absence of bar indicates that some species were not detected in that treatment group. The impact on the relative abundance (%) is summarized for some selected species: *S. mutans* (**c**), *L. casei* (**d**), *S. oralis* (**e**), *P. gingivalis* (**f**), *F. nucleatum* (**g**), and *S. salivarius* (**h**). Bars that do not share similar letters indicate statistically significant differences (*p* < 0.05), and the bars sharing similar letter(s) indicate no significant difference (*p* > 0.05). In legends, CHX0.1 and CHX0.2: Chlorhexidine 0.1 and 0.2%, respectively; ISOWE120: 120 mg/mL ISOWE.

**Table 1 biomolecules-13-01607-t001:** Relative abundance (%) of the taxonomical kingdom in each sample group.

Experimental Group	Unassigned	k__Archaea	k__Bacteria
Biofilm without sucrose	0.07	0.01	99.92
Biofilm with sucrose	0.00	0.04	99.96
CHX0.1	0.00	0.00	100.00
CHX0.2	0.00	0.00	100.00
ISOWE120	0.00	0.03	99.97
CHX0.1 + ISOWE120	0.00	0.00	100.00

**Table 2 biomolecules-13-01607-t002:** Relative abundance (%) of the unassigned group in each bacterial taxonomical level in each sample group.

Experimental Group	Phylum	Class	Order	Family	Genus	Species
Biofilm without sucrose	0.00	0.03	0.02	0.75	0.77	1.96
Biofilm with sucrose	0.10	0.06	0.01	0.39	1.62	2.07
CHX0.1	0.00	0.00	0.01	0.14	1.01	1.51
CHX0.2	0.00	0.00	0.00	0.06	1.38	1.08
ISOWE120	0.08	0.07	0.00	0.27	1.25	2.93
CHX0.1 + ISOWE120	0.00	0.01	0.01	0.14	0.94	1.73

CHX0.1 and CHX0.2: CHX 0.1 and 2%, respectively; ISOWE120: 120 mg/mL ISOWE; and CHX0.1 + ISOWE120: CHX 0.1% with ISOWE 120 mg/mL.

## Data Availability

The sequence data that support the findings of this study are openly available in https://figshare.com/account/home#/projects/179923 (accessed on 3 October 2023).

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
