# Peer review of "Sweet Orange Juice Processing By-Product Extracts: A Caries Management Alternative to Chlorhexidine"

_biomolecules, 2023, doi:10.3390/biom13111607_

Round 1
Reviewer 1 Report
Comments and Suggestions for Authors
This is a solid paper and the data are presented clearly. In materials and methods two questions arise. For the cytotoxic assays cells are treated with CHX or ISOWE prepared in completed DMEM. Does complete DMEM contain penicillin and streptomycin? Do these two components not interfere with CHX and ISOWE?
2.4.2 The biofil-growth medium was prepared bu diluting artificial saliva with basal media. What is the composition of artificial saliva and please be more specific about 'basal media'.
Author Response
This is a solid paper and the data are presented clearly. In materials and methods two questions arise. For the cytotoxic assays cells are treated with CHX or ISOWE prepared in completed DMEM. Does complete DMEM contain penicillin and streptomycin? Do these two components not interfere with CHX and ISOWE?
Penicillin and streptomycin were not used in the treatment solution preparations.
2.4.2 The biofilm-growth medium was prepared but diluting artificial saliva with basal media. What is the composition of artificial saliva and please be more specific about 'basal media'.
We are grateful to the reviewer for pointing out this omission. We have added the following information to the manuscript:
The artificial saliva used in the biofilm-growth medium (BGM) was a modified version of the artificial saliva described in Saha et al. (2023).
For information, see link: https://www.mdpi.com/2076-0817/12/5/657), the artificial saliva consisted of the following components (in g/L): porcine gastric mucin (2.5), NaCl (0.381), KCl (1.114), KH2 PO4 (0.738), ascorbic acid (0.002) as well as urea (9 mM) and L-arginine (5 mM). The basal media consisted of (g/L) protease peptone (10.0), tryptose peptone (5.0), yeast extract (5.0), L-cysteine hydrochloride (0.5), haemin (0.0002), and menadione (0.00004). Sucrose (2%) was added to the biofilm growth medium (BGM) to prepare a sucrose-fortified BGM (SFBGM).
Reviewer 2 Report
Comments and Suggestions for Authors
1- Lines 28 Lin, H. and Peddada [1], delete H after the name and rewrite as Lin and Peddada [1].
2- Line 83 It must start with a small letter.
3- The authors must include notes about the constituents of sweet orange juice extract as well as the pH and sugar content.
4- Lines 433- and 434-words sinensis and C. limon must be rewritten in italic form.
5- Line 446: words Citrus limetta and Citrus sinensis must be rewritten in italic form.
6- Line 475: Archaea must start with a small letter.
Author Response
- Lines 28 Lin, H. and Peddada [1], delete H after the name and rewrite as Lin and Peddada [1].
Corrected.
- Line 83 It must start with a small letter.
We checked the sentence on line 83 and confirmed that the formatting and grammar are correct. However, we apologize if we did not understand your meaning here.
- The authors must include notes about the constituents of sweet orange juice extract as well as the pH and sugar content.
We appreciate this comment and have clarified in the text, that in this study, we did not measure the pH and sugar content of the extract from sweet orange juicing by-products. However, detailed characterisation and composition of orange peel extracts has been reported in our previous study (https://www.mdpi.com/2076-0817/12/5/657).
- Lines 433- and 434-words sinensis and C. limon must be rewritten in italic form.
Corrected.
5- Line 446: words Citrus limetta and Citrus sinensis must be rewritten in italic form.
Corrected.
6- Line 475: Archaea must start with a small letter.
Corrected throughout the manuscript.
Reviewer 3 Report
Comments and Suggestions for Authors
This study evaluates the potential of flavonoid-rich industrial sweet orange waste extract (ISOWE) as an alternative to chlorhexidine (CHX) for dental caries management. Authors found that the ISOWE demonstrated no overall toxicity on buccal epithelial cells, and in combination with CHX significantly increased cell proliferation, whereas CHX alone considerably reduced cell viability and proliferation. Furthermore, ISOWE exhibited lower antimicrobial efficacy than CHX, however it was more active against pathogenic species while preserving the oral microbiome’s balance. In addition, exposure of cariogenic biofilm to the combination of ISOWE with 0.1% CHX produced similar efficacy to 0.2% CHX treatment alone. It is important to note that the current investigation is based on the previous study by Saha et al. (Antibiofilm Efficacies of Flavonoid-Rich Sweet Orange Waste Extract against Dual-Species Biofilms. Pathogens. 2023 Apr 28;12(5):657. https://doi.org/10.3390/pathogens12050657), in which the researchers reported the relevant results of ISOWE for the antibacterial effect on Streptococcus mutans and Lactobacillus casei, as well as for its antibiofilm activity against biofilms of these cariogenic bacteria. In this context, the present research provides new data for the ISOWE about its lower toxicity and enhanced antibacterial selectivity for pathogenic species in a complex multi-species biofilms developed from human oral microbiome samples compared to CHX. Therefore, results of this study have scientific and practical importance for the future application of industrial sweet orange waste extract in dental caries prevention. However, the manuscript needs minor revisions, based on the below provided comments, in order to be accepted for publication in the journal “Biomolecules”.
My comments to the authors are as follows:
1. In the section of Introduction (page 1, line 40): the statement that chlorhexidine is used in dental caries treatment as the “gold standard” should be changed because chlorhexidine is mostly used for the treatment of gingivitis and periodontitis. However, chlorhexidine is applied for the prevention of dental caries.
2. In the section of Results (page 10, line 375): “Figure 6” should be renamed to “Figure 5”.
3. Please provide the sources of funding at the end of the manuscript.
4. If applicable, please provide acknowledgments at the end of the manuscript.
5. Please format References (pages 15-17) according to the accepted style of the journal “Biomolecules”: list all surnames of authors for each article and do not italicize titles of the provided articles.
Author Response
- In the section of Introduction (page 1, line 40): the statement that chlorhexidine is used in dental caries treatment as the “gold standard” should be changed because chlorhexidine is mostly used for the treatment of gingivitis and periodontitis. However, chlorhexidine is applied for the prevention of dental caries.
Corrected.
- In the section of Results (page 10, line 375): “Figure 6” should be renamed to “Figure 5”.
The figure number in the respective figure was correct, which is Fig. 6. However, in the result section, it was mistakenly mentioned as Figure 5, which has been changed accordingly.
- Please provide the sources of funding at the end of the manuscript.
There was no specific funding received for the research.
- If applicable, please provide acknowledgments at the end of the manuscript.
Not applicable.
- Please format References (pages 15-17) according to the accepted style of the journal “Biomolecules”: list all surnames of authors for each article and do not italicize titles of the provided articles.
Done.